# Ginsenoside and Its Therapeutic Potential for Cognitive Impairment

**DOI:** 10.3390/biom12091310

**Published:** 2022-09-16

**Authors:** Hui Feng, Mei Xue, Hao Deng, Shiqi Cheng, Yue Hu, Chunxiang Zhou

**Affiliations:** 1School of Chinese Medicine, Nanjing University of Chinese Medicine, Nanjing 210024, China; 2School of Integrated Chinese and Western Medicine, Nanjing University of Chinese Medicine, Nanjing 210024, China; 3Tianjin Key Laboratory of Translational Research of TCM Prescription and Syndrome, First Teaching Hospital of Tianjin University of Traditional Chinese Medicine, Tianjin 300073, China; 4Department of Neurology, The Second Affiliated Hospital of Nanchang University, Nanchang 330008, China

**Keywords:** ginsenosides, cognitive impairment, pharmacological properties, apoptosis, inflammation

## Abstract

Cognitive impairment (CI) is one of the major clinical features of many neurodegenerative diseases. It can be aging-related or even appear in non-central nerve system (CNS) diseases. CI has a wide spectrum that ranges from the cognitive complaint with normal screening tests to mild CI and, at its end, dementia. Ginsenosides, agents extracted from a key Chinese herbal medicine (ginseng), show great promise as a new therapeutic option for treating CI. This review covered both clinical trials and preclinical studies to summarize the possible mechanisms of how ginsenosides affect CI in different diseases. It shows that ginsenosides can modulate signaling pathways associated with oxidative stress, apoptosis, inflammation, synaptic plasticity, and neurogenesis. The involved signaling pathways mainly include the PI3K/Akt, CREB/BDNF, Keap1/Nrf2 signaling, and NF-κB/NLRP3 inflammasome pathways. We hope to provide a theoretical basis for the treatment of CI for related diseases by ginsenosides.

## 1. Introduction

Cognitive impairment (CI) is a syndrome characterized by progressive degeneration of memory, learning, execution, attention, calculation, understanding, or judgment. Among the aging population, the incidence of CI is increasing worldwide. Approximately 10–15% of the population over the age of 65 develop CI, and more than half of them further progress to dementia within 5 years [1,2]. A study showed that about 46.8 million people had dementia in 2015, a number expected to increase to 131.5 million by 2050, which could seriously impact people’s elderly quality of life and longevity [3]. CI has become a major cause of the increased risk of mortality among the elderly and put a heavy economic burden on the healthcare system [4]. There are several etiologies of CI, including neurodegenerative diseases (e.g., Alzheimer’s disease (AD)), neurovascular diseases (e.g., strokes), psychiatric causes (e.g., depression), and non-central nerve system (CNS) diseases (e.g., diabetes) [5]. At present, drugs approved by the Food and Drug Administration for AD dementia, such as acetylcholinesterase inhibitors, MDA receptor antagonists, and dopamine-blocking agents, can provide symptomatic relief of CI [6]. However, the risk of the adverse effects of these drugs has gradually appeared [7]. The rate of gastrointestinal effects was reported to be up to 50% in a randomized trial [8]. In addition, aside from AD, few other dementia etiologies have approved drugs for cognitive symptoms [6]. Thus, there is always an urgent need for the investigation of safe and effective therapeutic candidates for CI. Recently, traditional herbal medicine has attracted widespread attention as a safe therapy candidate for CI.

Ginseng, the root of *Panax ginseng* Meyer C.A., Araliaceae (also called Asian ginseng), is a perennial herb widely used in traditional Chinese medicine (TCM) mainly distributed in China, Korea, and Russia, and it was also dubbed the “the king of herbs”. The name “ginseng” is also applied to other herbal plants, e.g., American ginseng (*Panax quinquefolius* L.) and notoginseng (*Panax notoginseng* (Burk.) Chen F.H.) [9]. The earliest records on the efficacy of ginseng can be traced back to the “Sheng-Nong-Ben-Cao-Jing” in the Eastern Han Dynasty in China about two thousand years ago, which recorded that “Ginseng can invigorate the five internal organs, calm the mind, stop palpitation, benefit intelligence, and prolong human longevity with a long-term administration”. Taking these as theoretical references, later generations of TCM physicians used ginseng for human healthcare for treating insomnia, amnesia, and dementia. Ginseng is classified into radix ginseng, red ginseng, white ginseng, and sugar ginseng based on the preparation method (steaming, fermentation, drying, etc.) [10]. Research has shown that ginseng exerts widely pharmacological effects on age- or disease-related CI and dementia, such as AD, Parkinson’s disease (PD), strokes, etc. [11,12,13].

Ginsenosides, the key active components of ginseng, are considered to be the main ingredients primarily liable for improving cognition without serious adverse effects [14,15]. Recently, the preclinical and clinical research on ginseng and ginsenosides in improving cognition has been rising. The present review specifically discussed the therapeutic potential of ginsenosides for CI and the potential mechanisms of action that have been globally conducted in the past few years, which is expected to provide a basis for further research and clinical application of ginseng and ginsenosides.

## 2. Classification, Chemical Structure, and Pharmacokinetics of Ginsenosides

Ginsenosides, a group of triterpenoid saponins, are the most widely studied ingredients of ginseng. The basic structure of ginsenosides consists of a sterane steroid core but with different glycosidic structures [16]. At present, there are more than 100 types of ginsenosides, which are classified into three main types according to the diverse glycosidic structures: the dammarane, oleanolic acid, and ocotillol types [16,17]. The dammarane type is a tetracyclic triterpenoid consisting of 17 carbon atoms arranged in a rigid steroidal backbone of four trans rings. According to a different site of glycosyl units, the dammarane type is further divided into two major groups: the 20(S)-protopanaxadiol (PPD) (e.g., ginsenosides Rb1, Rb2, Rc) and 20(S)-protopanaxatriol (PPT) types (e.g., ginsenosides Rg1, Rg2, Rh2) (Table 1) [18]. The PPD type is characterized by sugar linkages at C-3 and/or C-20, while the PPT type is characterized by sugar linkages at C-6 and/or C-20 [19]. Ro is the only typical representative ginsenoside of the oleanolic acid type characterized by C-3- and/or C-28 glycosyl chains [20]. Moreover, the ocotillo-type ginsenosides characterized by a five-membered epoxy ring attached to C-20 of the saponin moiety (e.g., pseudoginsenoside-F11 (PF11)) are the special ingredients of American ginseng (*Panax. quinquefolium* L.) distinguishing from *Panax ginseng* [21,22]. The structural characteristics of the main ginseng saponin types are summarized in Figure 1. Among these, dammarane-type ginsenosides (PPD and PPT types) occupy a dominant position in quantity and structural diversity [19]. Four PPD-type ginsenosides (Rb1, Rb2, Rc, and Rd) and two PPT-type ginsenosides (Re and Rg1) account for about 90% of the total saponins in raw *Panax. Ginseng* [23].

Different types of ginseng possess different contents of ginsenosides. A study showed that a total of 14 ginsenosides (Rg1, Re, Rf, Rh1, Rg2, Rb1, Rc, Rb2, Rb3, Rd, Rg3, Rk1, Rg5, and Rh2) were determined in Asian ginseng using high-performance liquid chromatography coupled with evaporative light scattering detection (HPLC-ELSD) [24], whereas in American ginseng, 43 ginsenosides were identified [25]. In addition, American ginseng has a different ginsenoside spectrum compared with Asian ginseng: (1) The top 3 major ginsenosides isolated from Asian ginseng are Rb1, Rg1, and Rb2, whereas the major content in American ginseng consists of Rb1, Re, and Rd. (2) PF11 only exists in American ginseng, whereas ginsenoside Rf is a special ingredient in Asian ginseng [26]. For notoginseng, it contains some of the same ginsenosides as Asian ginseng, e.g., Rb1, Rd, Re, Rg1, Rg2, and Rh1; however, no ocotillol- and oleanane-type ginsenosides can be found in notoginseng. In addition, some saponins are unique to notoginseng, e.g., notoginsenosides R1 and Rt [27,28].

In clinical application research, ginseng or ginsenosides are usually orally taken. However, 80% of the total ginsenosides are glycosylated ginsenosides (e.g., Rb1, Rb2, Rc, Rd, etc.), which have low solubility, low oral bioavailability in the gastrointestinal tract, and poor membrane permeability [17,29]. Deglycosylation, a reaction of cleaving the glycosyl from glycoproteins, is the most common metabolic pathway for ginsenosides in the gastrointestinal tract. An increasing amount of research has shown that ginsenosides can be extensively deglycosylated by intestinal bacteria (e.g., *Bacteroides* sp. and *Clostridium* sp.), which is vital for the pharmacological effects of ginsenosides [30]. Through deglycosylation, ginsenosides are transformed into different metabolites, such as Rh1, Rh2, and compound K, which present at a high bioavailability than their parental ginsenoside [31]. In a human pharmacokinetic study, nine ginsenosides (Rb1, Rb2, Rc, Rd, Rg3, CK, Rh2, PPD, and PPT) can be quantified in the human plasma after oral administration of red ginseng extracts for 14 days. Among them, the AUC and T_max_ value of PPD-type ginsenosides were higher than those of PPT-type ginsenosides [32]. The difference in pharmacokinetic features between PPD- and PPT-type ginsenosides is closely related to the number, type, and position of sugar chain linked to the sapogenin moiety [33]. As shown in Figure 2, the metabolic pathways of the PPD-type 20(S)-ginsenoside and PPT-type 20(S)-ginsenoside both have two deglycosylation processes in the intestine; the metabolic pathway represents deglycosylation at the C3, C6, or C20 position by β-glucosidase from intestinal microbiota [34]. Through stepwise deglycosylation, the ginsenoside Rd is transformed into F2 and Rg3, whereas Rg1, Re, and Rf are transformed into F1 and Rh1, respectively. Moreover, F2 and Rg3 further become compound K and Rh2, respectively. The final deglycosylated metabolites are presented as 20(S)-PPD and 20(S)-PPT [34].

Several studies have reported the pharmacokinetics of ginsenosides in brain tissue. One study showed that ginsenosides Rb1, Rb2, Rc, Rd, Re, Rf, Rg1, Rg3, and Ro were rapidly transported into the brain at 5 min after intravenous administration of ShenMai injection, a TCM preparation mainly composed of ginseng extract [35]. In addition, ginsenosides Rb1, Rg1, Ro, and Re can be detected in rat brain tissue after oral administration of Jia-Wei-Qi-Fu-Yin, a TCM decoction with ginseng as the main gradient. These data indicate that ginsenosides can penetrate through the blood–brain barrier (BBB) [36]. However, another study found that the average brain concentration of ginsenosides Rb1, Rg1, and Re was eight to 15 times lower than the corresponding content in plasma after the oral administration of ginseng extract in rats, which indicated that they have poor permeability to the BBB [37]. There are several studies focused on the distribution of ginsenosides in different regions of brain tissue. The imaging result of the brain implied that Rg1 might be distributed in the pons and medulla oblongata region of the brain at 15 min after intravenous administration in rats [38]. In addition, Re can be rapidly distributed into the cerebrospinal fluid and showed linear pharmacokinetics after subcutaneous injection in rats [39]. Moreover, ginsenosides Rg1, Rb1, Re, and Rd can be detected in the hippocampus, hypothalamus, olfactory bulb, striatum, cortex, and medulla oblongata of rats after administration of *Panax notoginseng* saponins through the nostril [40]. However, the knowledge of BBB permeability and distribution of ginsenosides in brain tissue remain limited; a better understanding of the pharmacokinetics of ginsenosides in the brain might substantially contribute to further research of their functions and mechanisms.

## 3. Protective Effect of Ginsenosides on CI

### 3.1. Neurodegenerative Diseases

Within the current decade, increased life expectancies greatly extended the number of elderly individuals who suffered from neurodegenerative disorders such as AD and PD [41]. Neurodegenerative diseases are characterized by the progressive loss of vulnerable populations of neurons, which leads to a cognitive decline in lifespan [42]. Accumulating evidence has suggested that ginseng, especially ginsenosides, has a protective effect on neurodegenerative-related CI in both clinical and preclinical studies.

For AD, the pathological hallmarks of this disease are proteinaceous deposits of amyloid β (Aβ), which can cause the syndrome of cognitive and functional decline [43]. Several pharmacological approaches have been widely used for intervening in AD-related CI, e.g., cholinesterase inhibitors and memantine. Meanwhile, ginseng and its main extract ginsenosides already show their protective effect on AD-associated CI in a few well-designed randomized controlled trials [11,44,45,46,47] (Table 2). A series of clinical studies were conducted by Heo [11]. They found that the ginsenoside complex, which is extracted from red ginseng, could improve patients’ cognitive function after at least 12 weeks of treatment. In another randomized controlled trial, 97 individuals were orally treated with *Panax ginseng* powder (which contains a total of 8.19% of ginsenosides) at a dose of 4.5 g/d or 9 g/day; the results demonstrated that ginsenosides supplement can increase the Alzheimer’s Disease Assessment Scale (ADAS) and Mini-Mental State Examination (MMSE) score, which revealed the improvement of cognitive function [45]. Furthermore, in a randomized, double-blind, placebo-controlled study, it was demonstrated that Korean red ginseng (KRG) administration at a dose of 1000 mg/day for 8 weeks can increase the gray matter volume and composite cognitive scores of healthy individuals [46].

Despite these clinical studies, numerous preclinical studies suggested vinous mechanisms for how ginsenosides exert their functions [52] (Table 3). For the components of ginsenosides, Rg1 and Rb1 are the major candidates that exert crucial neuroprotective effects to improve cognitive functions [53]. Studies have clarified that Rg1 can directly decrease the p-Tau level, Aβ generation, and amyloid precursor protein (APP) expression; increased the content of the brain-derived neurotrophic factor (BDNF); and attenuated hippocampal histopathological abnormalities to improve the cognitive capability in AD rodent models [53,54,55,56,57,58,59]. Moreover, indirectly, Rg1 can affect microbiota and change the abundance of gut microbiota to improve AD [60,61]. In addition, other ginsenosides including Rb1, Rg2, Rg3, F1, and compound K also showed anti-AD effects. Several studies suggested that Rb1 can inhibit apoptosis, decrease the Aβ level, reduce tau phosphorylation, attenuate inflammation, or even reduce insulin resistance to prevent AD cognitive deficit [62,63,64]. Compound K was able to induce antioxidant enzymes, attenuate cytotoxicity, and protect mitochondrial damage to reduce memory impairment in the AD model [65].

At present, to the best of our knowledge, no clinical study has reported the effect of ginsenosides on PD, which is characterized by the progressive death of dopamine neurons. However, in a 1-methyl-4-phenyl-1,2,3,6-tetrahydropyridine (MPTP) mouse model of PD, Rb1 was confirmed to bind with GABAARα1 and increase its expression, thus promoting the prefrontal cortical γ-aminobutyric acid (GABA) level and GABAergic transmission to reduce MPTP-induced dysfunctional gait dynamic and CI [66]. Moreover, Rb1 and Rg1 together reversed the MPTP-induced cell death and improved performance in a passive avoidance-learning paradigm, suggesting that these two ginsenosides have neurotrophic and selective neuroprotective properties and may contribute to enhancing cognitive deficit in PD [67].

### 3.2. Neurovascular Diseases and Other CNS Diseases

Neurovascular diseases, mainly including ischemic and hemorrhagic stroke, can lead to vascular dementias and other forms of neurological dysfunction and degeneration in the long term [78]. Ginsenosides act as an antistroke factor in the progression of stroke and stroke-related dementias. In a photothrombotic stroke model, Rb1 was found to alleviate the morphological lesion and promote cognitive and sensorimotor deficits [68]. Ginsenoside Rd was able to attenuate ischemia-induced enhancement of tau phosphorylation, leading to the amelioration of the behavior impairment [58]. In parallel, ginsenoside Rd also plays a key role in chronic cerebral hypoperfusion-induced CI by upregulating the BDNF [74].

For other CNS diseases that may cause CI, ginsenosides also show neuroprotective functions. In a chemotherapy-induced CI mouse model, Rg1 was given 1 week prior to model conduction. The results demonstrated that Rg1 treatment ameliorated cortical neuronal dendritic spine elimination and improved chemobrain-like behavior; moreover, Rg1 also affected the expression of cytokine mediators and microglial polarization, indicating that Rg1 can exert an anti-inflammation effect and promote neuroplasticity in certain brain regions that are linked with cognition [79]. In a traumatic brain injury rat model, ginsenosides showed anti-neuroinflammation and antioxidative stress properties and may inhibit the microglial pathway against head-trauma-induced CI [80].

### 3.3. Psychiatric Disorders

Psychiatric disorders, such as anxiety, depression, and schizophrenia, usually with prolonged stress, may cause extensive loss of neurons leading to deficits in cognitive performance. Numerous studies have focused on the pharmacological effect of ginsenosides on psychiatric disorders caused by CI. However, most of the current research involves preclinical studies in vivo or in vitro. Up to now, only a few clinical researchers reported the effectiveness of ginsenosides on psychiatric disorders. By using a randomized double-blind placebo-controlled design, Chen et al. found that patients with schizophrenia have improved visual working memory and reduced extrapyramidal symptoms after treatment with HT1001 (i.e., a proprietary North American ginseng extract containing known levels of active ginsenosides) [48]. Another 6-week, double-blind, randomized, placebo-controlled trial demonstrated that an extract of Korean red ginseng (KRG) might help to stabilize the sympathetic nervous system for people with high-stress occupations [49] (Table 2).

For preclinical studies, the chronic restraint stress model is a well-known psychosocial stress model to mimic the environment-induced CI for humans. There are already several studies that investigated the role of ginsenosides on the stress model. Administration of ginsenoside Rd has been reported to increase the antioxidant enzyme activities in the hippocampus; meanwhile, serum inflammation factors can be reduced [75]. In the thermal stress HT22 cell model, ginsenoside Rg5 has been confirmed to prevent cell apoptosis and alter the expression of CI-associated genes in vitro [76]. With treatment of American ginseng, mice underwent chronic unpredictable stress and had better performance in a behavioral test; moreover, mitochondrial enzyme complex activities, oxidative stress markers, and expression of proinflammatory cytokines were altered [81].

### 3.4. Non-CNS Diseases

CI not only is tightly associated with CNS diseases but also can appear in systemic or metabolic diseases, e.g., diabetes, heart failure, cancers, and alcoholism. Particularly, ginsenosides Rb1 and Rg1 have been reported to reduce cognitive deficits through vinous mechanisms in non-CNS diseases. Scientific evidence has revealed that ginsenoside Rb1 improves CI in the insulin resistance model and high glucose-induced model, suggesting that Rb1 can ameliorate diabetic-induced CI [69,70]. Moreover, ginsenoside Re has also been confirmed to attenuate diabetes-associated cognitive deficits through antioxidative stress and anti-inflammation properties [77]. In the alcoholism model, Rg1 was able to alleviate cognitive deficits by reducing neuro-excitotoxicity [71].

With natural aging, the brain undergoes progressive behavioral retrogression with cognitive and motor declines. Ginsenosides can be a useful adjuvant treatment against neurodegeneration-caused cognitive decline. Even among healthy adult individuals, ginsenosides are reported to enhance working memory and cognitive performance in their daily work (Table 2) [50,51]. The anti-aging effect of ginsenoside Rg1 was tested in an aging rodent model induced by d-galactose or d-galactose and AlCl3 [72,73]. Results demonstrated that Rg1 effectively improved the cognitive performance of experimental animals by protecting neural stem cells and progenitor cells, reducing astrocytes activation, and inhibiting neural apoptosis [72,73].

To conclude, multiple ginsenosides are able to ameliorate CI in different diseases. Among them, which ones may play a stronger role against CI? Research has demonstrated that a decrease in the number of sugar moieties of ginsenosides was shown to weaken their anti-CI effects. Several studies have compared the effects and mechanisms on improving CI of Rb1 and Rg1, which have been proved to be high in content of *Panax ginseng*. In scopolamine-induced mice, both Rg1 and Rb1 intraperitoneal administration at 6 and 12 mg/kg improved the CI in mice and inhibited the decrease in 5-HT induced by scopolamine. However, Rb1, with four sugars, was more effective than ginsenoside Rg1, which contains only two sugars [82]. Similar results have been shown in Yang et al.’s study [53]. Moreover, another study reported that both Rg1 and Rg2 by intraperitoneal injection at 30 mg/kg/day improved spatial learning and memory deficit in APP/PS1 mice, but the effect of Rg1 was more obvious [83]. PF11, an ocotillol-type ginsenoside, whose mother nucleus structure is similar to that of dammarane ginsenosides, also has a strong effect on the prevention of CI, especially in the treatment of AD-related CI [84]. Interestingly, compound K, a metabolite of PPD-type ginsenosides Rb1, Rb2, and Rc, seems more bioavailable than other ginsenosides such as Rd [85]. However, at present, there are relatively few studies on the comparison of the different effects of various ginsenosides on preventing CI, which may be an important perspective of future studies. Moreover, many other bioactive compounds from herbal plants that may contribute to the prevention of CI [86] are listed in Table 4.

In spite of the strong anti-CI effect of ginsenosides in CNS or non-CNS diseases, side effects may occur especially with long-term use, called “ginseng abuse syndrome”. The most common side effects of ginsenosides include diarrhea, vomiting, hypertension, skin rash, insomnia, breast pain, and vaginal bleeding [87]. In a two-year human study, 14 to 26 of 133 participants showed symptoms of hypertension, euphoria, irritability, insomnia, edema, and diarrhea after taking ginseng for a long time. However, the validity of these studies is difficult to assess because of the absence of a control group and the fact that the subjects used different ginseng preparations with the dosage ranging from 1 to 30 g per day and were not controlled for other bioactive substances intake (e.g., caffeine) [88]. Because one of the main side effect of ginseng is hypertension, it is highly recommended that patients should discontinue ginseng use at least 7 days before surgery to reduce perioperative morbidity related to the herbal supplements [89,90]. In addition, several studies also reported that ginseng may cause breast pain and vaginal bleeding in postmenopausal women in a few cases, which might be related to the physiological estrogen-like effect of ginseng [91,92]. Therefore, new administration routes related to novel materials (e.g., nanoparticles) that may potentially reduce the side effect of ginseng or ginsenosides treatment are urgently needed.

## 4. Pharmacological Properties of Ginsenosides of CI

### 4.1. Regulating Cholinergic Transmission

Cholinergic transmission occupies a key role in cognitive function. Acetylcholine (Ach) is an important neurotransmitter in cholinergic transmission. However, the excess Ach is broken down by the enzyme acetyl cholinesterase (AChE) into choline and acetate. Thus, overexpression of AChE can cause CI via interfering synaptic transmission and neuroplasticity [102]. Several studies have shown that cholinergic dysfunction is closely related to cognitive decline in AD and PD [103,104]. In an in vitro enzymatic assay, ginsenosides, including Rb1, Rb2, Rc, Rd, Re, Rg1, Rg3, F1, Rk3, and F2, have been confirmed to show a strong inhibitory effect on AChE activity [105,106]. Pharmacokinetic studies indicated that in AD models, Rg1 and Re were rapidly transported into the hippocampus after intraperitoneal injection, and the extracellular level of Ach was significantly increased [39,107]. In a scopolamine-induced mice model, ginsenoside Rh3 reduced AChE activity in a dose-dependent manner and protected memory deficit [108]. Ginsenosides Rb1, Rg1, Rg5, Rk1, and 20(S)-PPT have similar effect as Rg3. Studies demonstrated that they exerted a memory-improving effect via the inhibition of AChE activity attributing to increased Ach levels in different regions of the hippocampus [82,109,110]. Moreover, Re and Rd orally administered rapidly transported into brain tissue and induced the expression of acetyltransferase and the vesicular Ach transporter, which are required for cholinergic neurotransmission, as well as the increased level of Ach in Neuro-2a cells [111]. Rb1 also improved CI induced by cisplatin via increasing the cholineacetyltransferase activity and improving cholinergic system reactivity in the hippocampus [112]. In addition, ginsenoside Re also regulated AChE activity and increased the Ach level via regulating the c-Jun N-terminal protein kinase pathway in high fat diet induced mouse brains [113]. Furthermore, in an LPS-induced CI model, ginsenoside Rg1 administration also increases the alpha7 nicotinic Ach receptor expression in the prefrontal cortex and hippocampus [114].

### 4.2. Inhibiting Oxidative Stress

With age- and disease-dependent loss of mitochondrial function, the production of reactive oxygen species (ROS) is increasing. Excessive ROS in the brain affects the calcium homeostasis and nuclear and mitochondrial DNA, and further causes alteration of neurotransmission and synaptic activity. Moreover, altered metabolism caused by the imbalance of ROS can accelerate the accumulation of Aβ and hyperphosphorylated Tau protein, leading to cognitive dysfunction [115]. Numerous studies have clarified that ginsenosides, including Rg1, Rb1, PF11, Rd, Rh1, Rh2, Re, and PPT, have an antioxidative function in disease-mediated CI [110,116,117,118,119,120,121,122,123]. The antioxidative effect of ginsenosides is represented as the reduction of biochemical markers regarding oxidative stress, e.g., superoxide dismutase (SOD) levels, NADPH oxidase 2 (NOX2) expression, glutathione peroxidase, and ROS. Specifically, studies have reported that Rg1 can mitigate the elevated levels of ROS, decrease NOX2 expression, and increase the activity of SOD and glutathione peroxidase, thus decreasing oxidative stress in the brain [59,116]. Rb1, PF11, PPT, and Rh2 have a similar effect as Rg1; studies have demonstrated that they exert an antioxidative effect by increasing the level of SOD, catalase, and total glutathione while lowering the level of malondialdehyde and H_2_O_2_, thus alleviating mitochondrial damage and ROS production [110,117,118,119,120,121]. However, Rd only reported increased antioxidant enzyme activities in the hippocampus region [122,123].

### 4.3. Protecting against Apoptosis

Neuronal loss is a critical pathological substrate of cortical atrophy, which strongly correlates with cognitive deficits [124]. The mechanisms of neuronal death are vinous in different diseases. Studies have reported that ginsenosides may have a positive effect on neuronal loss and neuronal apoptosis. Among ginsenosides, Rg1, Rb1, and Rg5 showed a strong anti-apoptosis effect on the cognitive impaired model. For instance, pretreatment of Rg1 and Rb1 effectively rescued neuronal apoptosis markers, e.g., cleaved caspase-3 and Bax. However, the expression of Bcl-2 can be upregulated [84]. In aging-induced cognitive decline, Rg1 incited apoptosis in the hippocampus and prefrontal cortex by promoting the expression of anti-apoptotic protein Bcl-2 and enzyme cleaved-caspase3; moreover, the expression decline of fibroblast growth factor 2 (FGF2) and BDNF can be restored by Rg1 treatment [125]. In addition, daily administration of Rb1 for 2 weeks in an Aβ_1–40_-induced AD model showed a strong anti-apoptotic effect by reducing the levels of Bax and cleaved caspase-3 while upregulating the level of Bcl-2 in the hippocampus [62]. For Rg5, one study has investigated its anti-apoptotic effect on a thermal-stress-exposed HT22 cells model. The results demonstrated that Rg5 may act as an anti-apoptotic factor that can prevent hippocampal cell damage by retaining the p21 expression and suppressing the PARP cleavage [76]. To sum up, ginsenosides protect against cell apoptosis by regulating the key proteins of the Bcl-2 family and caspase family, e.g., Bcl-2, Bax, and cleaved caspase-3, in a caspase-dependent apoptotic process.

### 4.4. Inhibiting Inflammation

In patients with CI, although they have various pathological changes, the levels of several inflammatory mediators are increased not only in the plasma but also in the cerebrospinal fluid, e.g., interleukin-6 (IL-6), C-reactive protein (CRP), tumor necrosis factor-α (TNF-α), and IL-1β [125,126,127]. Moreover, the innate immune cells (e.g., neutrophils, monocytes, T cells, and microglia) can be activated in an inflammatory state, resulting in neuronal apoptosis and damaged neuronal connections [128]. Chronic low-grade inflammation has been confirmed to be closely associated with CI in many diseases. For instance, in AD, the activation of microglia accelerates the release of proinflammatory cytokines and neurotoxins, leading to the accumulation of Aβ and neurodegeneration [129]. Several studies already reported the effects of ginsenosides on inflammation that influence cognitive conditions. Ginsenoside Rg1 was confirmed to have an anti-inflammatory effect on an animal model of chemotherapy-induced CI by inhibiting the elevated level of proinflammatory cytokines (TNF-α, IL-6) while promoting the levels of anti-inflammatory cytokines (IL-4, IL-10) [79]. Moreover, Rg1 also attenuated the microglial polarization from M2 to M1 phenotypes [77]. In another study, Rg1 and Rb1 showed a protective effect in inhibiting the activation of astrocytes and microglia in the hippocampus of an Alzheimer’s mice model [53]. Furthermore, Rb1 treatment can inhibit the expression of ASC and caspase-1 in the hippocampus of mice and maintain microglial homeostasis [53,130]. In LPS-induced systemic inflammation caused by CI, ginsenoside Rg1 treatment improved spatial learning and memory by enhancing the expression of the α7nACh receptor, which is a key regulator in inflammation [114]. Not only in the pathological animal model but also in the aging model, Rg1 also showed an anti-inflammatory effect for it decreased the level of proinflammatory cytokines (IL-1β, IL-6, and TNF-α) and inhibited astrocytes activation via decreasing the level of Aeg-1 expression [72]. Aside from Rg1 and Rb1, other ginsenosides (e.g., Rf, Rg5, and compound K) also play an important role in disease-induced CI by attenuating neuroinflammation. Studies have reported that ginsenoside Rf could reduce the level of interferon-gamma (IFN-γ), thus alleviating the Aβ-induced inflammation reaction [131]. In the streptozotocin (STZ)-induced memory-impaired model, Rg5 treatment showed great effectiveness for improving the STZ-induced CI; meanwhile, it significantly decreased the levels of inflammatory cytokines TNF-α and IL-1β [132]. In addition, compound K was also confirmed to suppress the inflammatory response by targeting the nod-like receptor pyrin domain containing 3 (NLRP3) inflammasome pathway to improve the cognitive dysfunction in a diabetic mice model [133]. The above research showed that ginsenosides Rg1, Rb1, Rf, Rg5, and compound K mainly exert their function to improve CI by regulating the activation of astrocytes and microglia, decreasing the level of proinflammatory cytokines, and modulating the expression of key factors in the NLRP3 inflammasome pathway.

### 4.5. Enhancing Synaptic Plasticity and Neurogenesis

Synapses are special structures in neurons that are responsible for chemical or electrical signals passing from one cell to another. Neurotransmitters such as glutamate, GABA, and dopamine are released from the presynaptic neuron to a postsynaptic neuron. The impairments of neuroplasticity such as synapse loss and neuronal atrophy in the prefrontal cortex and hippocampus play a critical role in learning and memory dysfunction [134]. Soluble Aβ can negatively affect synaptic plasticity, damage synapses, and accelerate the accumulation of Aβ in the neurons [135]. Studies already reported the effect of ginsenoside on synaptic plasticity to enhance cognitive functions. Ginsenoside Rg1 treatment in different pathological models not only functionally increased the sensitivity of evoking PS and restored the long-term potentiation (LTP) but also mechanically upregulated the expression of synaptic plasticity-associated proteins, e.g., glutamate receptor 1 (GluR1), synaptophysin, postsynaptic density 95 (PSD95), calcium/calmodulin-dependent protein kinase II alpha, etc. [136,137,138,139,140,141,142,143]. In addition, Rg1 can regulate the expression of miR-134 to exert its effect on neuroplasticity [144]. Ginsenoside Rb1 has a similar effect as Rg1. Studies demonstrated that in an MPTP mouse model, treatment with Rb1 promoted LTP and glutamatergic and GABAergic transmission in the hippocampus of mice; the underlying mechanism was that Rb1 sequentially improved the expression of PSD95 and α-synuclein in hippocampal CA3 [66,145]. Rb1 also increased the density of a synaptic marker protein including synaptophysin PSD95 in a different pathological model such as chronic restraint stress as well as health animals [142,146]. Moreover, Rb1 administration promotes LTP and increased cell survival in the dentate gyrus and hippocampal CA3, suggesting that Rb1 enhanced not only synaptic plasticity but also neurogenesis [147,148]. However, there is still a lack of mechanism-related studies to explore the underlying mechanisms of the exact target of ginsenosides and how they work on synaptic plasticity and neurogenesis to promote cognitive function.

In conclusion, ginsenosides improve CI mainly through regulating cholinergic dysfunction, inhibiting oxidative stress, protecting against apoptosis, inhibiting inflammation, and enhancing synaptic plasticity and neurogenesis.

## 5. Signaling Pathways Involved in the Treatment of CI by Ginsenosides

The molecular mechanisms responsible for the pharmacological activities of ginsenosides peradventure are associated with their ability to block and/or activate mediators and transcription factors in several signaling pathways including the PI3K/Akt (Figure 3), CREB/BDNF (Figure 4), Keap1/Nrf2 (Figure 5), and NF-κB/NLRP3 inflammasome pathways (Figure 6).

### 5.1. PI3K/Akt Signaling Pathway

The phosphoinositide 3 kinase (PI3K)/Akt pathway, an intracellular signal transduction pathway, plays a wide range of regulatory effects on cellular metabolism, survival, and proliferation [149]; meanwhile, it exerts a critical role in neuronal survival and synaptic plasticity [150]. Various cytokines, such as insulin, insulin-like growth factor-1 (IGF-1) [151], and FGF2 [152], are the important upstream factors for PI3K activation, which promote phosphatidylinositol-3,4,5-bisphosphate to recruit phosphoinositide-dependent protein kinase (PDK) and Akt to the plasma membrane and further activate Akt via promoting PDK to phosphorylate Thr308 and Ser473 on Akt. Active phosphorylated Akt can regulate multiple downstream substrates, such as glycogen synthase kinase 3β (GSK3β), mammalian target of rapamycin (mTOR), and Bad, and further can exert neuroprotective effect [153]. Numerous studies have shown that ginsenosides Rg1 [73,116,139,154], Rg2 [155], Rb1 [68,119], Rd [58,156], and compound K [157] could upregulate the PI3K/Akt signaling pathway in age- or disease-related CI. Molecular mechanisms of ginsenosides-medicated activation of the PI3K/Akt pathway could be classified as two aspects.

#### 5.1.1. Activating PI3K/Akt Signaling Pathway by Upstream Cytokines

Several studies have confirmed that IGF-1-activated PI3K/Akt can mitigate cognitive dysfunction in high-fat-diet rats [158] and inhibit sevoflurane-induced activation of hippocampal cells in aged rats [159,160]. In clinical research, a lower plasma level of IGF-1 is confirmed as an important risk factor for poor cognition in PD [161] and developing dementia in AD [162]. Moreover, another study has shown that FGF2 transfer can improve learning deficit, reduce Aβ deposition, and enhance neurogenesis in a mouse model of AD while enhancing Aβ phagocytosis in primary cultured microglia [163]. Suppression of hippocampal neurogenesis in CI mice has been demonstrated to be closely associated with inhibiting the FGF2/PI3K/Akt signaling pathway [164,165]. In a streptozotocin-induced rat, ginsenoside Rg5 can significantly ameliorate learning and memory dysfunction and Aβ deposition in both the cerebral cortex and hippocampus via activating IGF-1 [132]. However, in vitro, ginsenoside Rg5 improved the lipopolysaccharide-induced overactivation of microglial cells, which is involved in the cognitive decline of neurodegenerative diseases, and exerted anti-inflammatory effects via inhibiting the PI3K/Akt signaling pathway [166]. Moreover, ginsenoside Rg1 can inhibit neuronal apoptosis in the hippocampus of aging mice via activating the FGF2/Akt pathway [73].

#### 5.1.2. Regulating PI3K/Akt Signaling Pathway and Downstream Molecules

GSK3β, a major Tau kinase, can promote tau phosphorylation and neurofibrillary tangles formation [167]. It has been demonstrated that GSK3β overexpression mediates tau hyperphosphorylation and neurodegeneration in mice, which causes learning decrease [168,169]. However, Akt can inhibit the GSK3β activity by phosphorylating it at serine residues 9 [157]. Activating the PI3K/Akt/GSK3β signaling pathway has been confirmed to be an effective intervention strategy for AD dementia [170]. In ischemic stroke rats, Rd decreased hippocampal cell loss, mitigated tau phosphorylation, and improved cognitive dysfunction, as well as inhibited the activity of GSK-3β and activated the PI3K/Akt signaling pathway, while an antagonist of PI3K eliminated the effect of Rd on GSK-3β activity and CI, suggesting that Rd improved CI via regulating the PI3K/Akt/GSK-3β signaling pathway [58]. In a vascular dementia rat model, ginsenoside compound K ameliorated CI, reduced the Aβ accumulation, and upregulated the levels of p-GSK3β (Ser9) and p-Akt in the hippocampus. Moreover, in aging rats induced by isoflurane anesthesia, an inhalation anesthetic frequently used in clinics, which can impair cognition [171], Rg1 was found to improve CI via increasing the levels of PI3K, AKT, and GSK-3β in the hippocampus [154]. Otherwise, Rg1 improved the tau hyperphosphorylation, reduced the deposition of Aβ_1–42_, and improved the cognitive deficit of the AD tree shrew via regulating the GSK-3β signaling pathway [172]. mTOR is confirmed to exert a significant effect on synaptic plasticity and β-amyloid clearance via regulating autophagy [173]. Huang et al. reported that ginsenoside Rg1 could increase the expression of p-PI3K, p-Akt, and p-mTOR and thus inhibit autophagy and autophagic injury in PC12 cells induced by oxygen-glucose deprivation and reoxygenation [174]. A PPD derivative, 1-(3,4-dimethoxyphenethyl)-3-(3-dehydroxyl-20(*S*)-protopanaxadiol-3b-yl)-urea could reduce senile plaque, protect synaptic integrity, and improve CI in a mouse model of PD through activating PI3K/Akt/mTOR-mediated autophagy [175]. In addition, alkaline hydrolyzed products of ginsenosides also exert a protective effect on CI in rats under a simulated long-duration spaceflight environment via activating the PI3K/Akt/mTOR signaling pathway [176].

In addition to all the above-mentioned findings, the PI3K/Akt signaling pathway also plays an important role in apoptosis. Akt activation can inhibit the phosphorylation of Bad, which induces apoptosis via inactivating Bcl-2 [177]. Gui et al. reported that ginsenoside Rg2 ameliorates neurotoxicity and memory impairment via inhibiting Akt-mediated apoptosis in an AD rat model; in vitro, Rg2 increased the Bcl-2/Bax ratio, attenuated the expression of active caspase-3, and thus inhibited apoptosis via increasing the phosphorylation of Akt in Aβ25-35-induced PC12 cells [155,178]. Ginsenoside Rb1 also inhibits the methylglyoxal-induced apoptosis in SH-SY5Y cells via activating the PI3K/Akt signaling pathway [119].

### 5.2. CREB/BDNF Signaling Pathway

cAMP-responsive element-binding protein (CREB), an important nuclear transcription factor, is a significant downstream mediator of Aβ toxicity implicating learning and memory [179]. CREB phosphorylation at Ser133 is a crucial step to activate CREB-dependent transcription [180]. Phosphorylated CREB can promote the transcription of BDNF widely expressed in the hippocampus and cerebral cortex [181], which is involved in synaptic plasticity [182], neuronal differentiation, survival [183], and memory formation [184]. It has been reported that in AD patients’ postmortem hippocampal tissues, the expression of p-CREB, CREB, and BDNF remarkably decreased; meanwhile, the level of Aβ_1–41_ increased by about 505% and showed a negative correlation with CREB protein expression [185]. An increasing amount of research has demonstrated that the CREB/BDNF signaling pathway is an important regulatory mechanism of improving age- or disease-related CI for ginsenosides [54,73,136,186,187]. In APP/PS1 mice, ginsenoside F1 was found to improve spatial working memory ability and decrease Aβ plaques via inhibiting the expression of p-CREB and the BDNF in the cerebral cognitive region [188]. In scopolamine-induced mice, both ginsenoside Rg5 and Rg3 can ameliorate CI, regulate AchE activity, and increase the expression of p-CREB and the BDNF in the hippocampus [108,132]. Moreover, okadaic acid-induced Tau hyperphosphorylation may be relieved by ginsenoside Rg1 pretreatment via inhibiting the BDNF [187]. However, it was reported that ginsenoside Rh1 could increase cell survival in the hippocampus via increasing the expression of the BDNF, and hippocampal cell proliferation was not affected [189].

B-type tropomyosin-related kinase B (TrkB), a high-affinity receptor for mature BDNF, is crucial to the protective functions of the BDNF on the central nervous system [184,190]. It has been demonstrated that BDNF–TrkB signaling is required for LTP induction, synaptic transmission, and dendritic growth [191]. The expression of the BDNF, p-TrkB, and TrkB decreases in both the hippocampus and cerebral cortex of different animal models of CI including AD, TBI, DM, etc. [191,192,193]. In ischemic stroke mice, PF11 preadministration improved CI and hippocampal atrophy; promoted neuronal generation and survival; and increased the expression of CREB, p-CREB, m-BDNF, and TrkB in the dentate gyrus and striatum, while these effects were eliminated by a specific inhibitor of TrkB in OGD/R-induced neural stem cells [194]. In the chronic restraint stress (CRS)-induced CI model, ginsenoside Rb1 could improve synaptic plasticity, inhibit apoptosis, and activate the BDNF/TrkB signaling pathway [146]. What is more, ginsenoside Rg1 repaired hippocampal LTP and memory, reduced Aβ_1–42_ deposition and tau hyperphosphorylation, and upregulated the expression of BDNF and p-TrkB in both AD and aging mice [73,136].

Multiple pathways including PI3K/AKT, cAMP-protein kinase A (PKA), Ca^2+^/calmodulin-dependent protein kinase II (CaMKII), and extracellular signal-regulated kinase (ERK) can promote the nuclear translocation and phosphorylation at Ser133 of CREB and thus activate the CREB/BDNF/TrkB signaling pathway [195,196,197]. In AD mice, ginsenoside Rg1 ameliorated the impairment of learning and memory; decreased the hippocampal Aβ content; and upregulated the expression of PKA, BDNF, p-CREB, and CREB in the hippocampus [53]. In addition, ginsenoside Rg1 performed significant cognitive protection via increasing the expression of p-ERK, BDNF, and TrkB in the prefrontal cortex of a CRS-induced CI model [186]. Ginsenoside Rg1 also activated CaMKIIα and increased the expression of p-ERK1/2 and p-CREB in PC12 cells, while these effects were partly abolished by the preincubation of a CaMKIIα inhibitor, indicating that ginsenoside Rg1 activates the ERK/CREB pathway by CaMKIIα [198]. In addition, ginsenoside Rd performed neuroprotection via activating the PI3K/CREB/BDNF/TrkB signaling pathway in the hippocampus of CRS-induced CI mice [52]. Besides these, long-term ginsenosides (including ginsenoside Rb1, Rb2, Rc, Rd, Re, Rf, Rg1, Rg2, Rg3, and Ro) could decrease neuron destruction; upregulate the expression of PSD-95 and NMDAR1, the plasticity-related proteins; and activate the PKA/CREB/BDNF signaling pathway in the hippocampus of aging and AD mice [199,200].

### 5.3. Keap1/Nrf2 Signaling Pathway

Nuclear factor E2-related factor 2 (Nrf2), a nuclear transcription factor, is an important regulator of intracellular redox homeostasis and also a key molecule for clearing ROS [201]. Kelch-like ECH-associated protein 1 (Keap1) exists in the cytoplasm by binding to Nrf2 and mediates the inhibition of Nrf2 [202]. As the level of intracellular ROS increases, Nrf2 dissociates from Keap1; transfers to the nucleus and combines with the antioxidant-response element (ARE); and further induces downstream antioxidative factors including heme oxygenase-1 (heme oxygenase 1, HO-1), NAD(P)H: quinone oxidoreductase 1 (NQO1), GSH, and SOD and thus exerts antioxidant activities [203]. It has been confirmed that the Nrf2 localization in the neuron nucleus in the hippocampus of AD patients significantly decreased [204]. In addition, the Nrf2 signaling pathway was also demonstrated to be closely associated with Aβ pathology, tauopathy, and synaptic plasticity impairment in animal models of CI [205,206,207]. Among the ginsenosides, Rb1, Rg1, Re, Rh2, and PF11 showed a positive effect on the Nrf2 signaling pathway in the CI model. For instance, Rb1 attenuated neuronal damage and CI via activating the Nrf2/HO-1 pathway in pentylenetetrazol (PTZ)-induced rats [208]. In the AD model, Rb1 exerted a protective effect on cognitive function; inhibited Keep-1 expression; and upregulated the expression of Nrf2, HO-1, GSH, and SOD in the hippocampus. Moreover, Rb1 and Rg1 pretreatment decreased ROS levels, induced SOD activation, and promoted the nucleus translocation of Nrf2 in rotenone-induced SH-SY5Y cells [121]. In another study, ginsenoside Re ameliorated CI and enhanced the expression of Nrf2, HO-1, and some synapse-associated proteins (synaptophysin, SYP, and PSD95) in the hippocampus of CRS-induced mice [209]. In addition, oral administration of PF11 could reverse the decrease of SOD, GSH, and Nrf2 in the hippocampus of D-gal-induced mice [118]. Red ginseng extract (including compound K, Rb1, Rg1, Rc, Rd, Rh2, etc.) could improve scopolamine-induced cognitive dysfunction and hippocampal damage, as well as upregulate the expression of Nrf2, HO-1, and NQO1 in the hippocampus of mice [210].

### 5.4. NF-κB/NLRP3 Inflammasome Pathway

Nuclear factor-κB (NF-κB) and NLRP3 inflammasome are two important drivers of inflammation. NF-κB is a transcription factor involved in the regulation of neuroinflammation and glial activation in the CNS. The overactivation of NF-κB promoted microglia into an inflammatory state, increased the diffusion of tau protein, and reduced spatial memory in AD mice [211]. It was confirmed that ginsenoside compound K can inhibit NF-κB p65 nuclear translocation in Aβ42 oligomers-induced BV2 cells [212]. NF-κB can further activate the NLRP3 inflammasome, a proteolytic complex present in various mammalian cells, which comprises the NLRP3 protein, the adaptor apoptosis-associated speck-like protein, and pro-caspase-1 [213]. NLRP3 inflammasome activation promotes the conversion of pro-caspase-1 to active caspase-1, which is crucial to the release of mature IL-1β and IL-18 [214,215]. It has been confirmed that NLRP3 inflammasome activation can induce tau hyperphosphorylation and synaptic plasticity deficit and inhibit Aβ clearance [216,217,218]. Several studies already reported the protective effects of ginsenosides on cognitive conditions via inhibiting the NLRP3 inflammasome pathway. Ginsenoside compound K showed the effects of improving cognitive deficit and inhibiting the expression of NLRP3, ASC, caspase-1, and mature IL-1β in the hippocampal tissues of diabetic mice [133]. Rb1 administration ameliorated Aβ deposition, inhibited the activation of microglia, and reduced the expression of ASC and caspase-1 in the hippocampus of AD mice [53]. In D gal-induced mice model, PF11 was confirmed to inhibit neuroinflammation via suppressing NLRP3 inflammasome activation in the hippocampus [133]. Another study showed that ginsenoside Re could improve CI and decrease the expression of NLRP3, ASC, caspase-1, IL-1β, and IL-18 in the hippocampus of CRS-induced mice [209]. Moreover, ginsenoside Rf decreased the tau phosphorylation and active caspase-1 expression in Aβ-induced N2A cells [131]. In addition, IL-1β has strong proinflammatory properties and further activates some other inflammatory factors including TNF-α and IL-6 and induces an inflammatory cascade [219,220], which is also closely related with CI in many diseases. Among ginsenosides, Rg1, Rg3, and Rg5 have been demonstrated to improve CI via inhibiting the expression of IL-1β, TNF-α, and IL-6 in the hippocampus of an aging, STZ-induced, or LPS-induced animal model [72,132,221].

## 6. Conclusions

CI has been linked to many diseases and caused a great burden to the whole society; thus, novel therapeutic strategies are urgently needed. As a group of triterpenoid saponins, ginsenosides exert protective effects on CI in diverse disease models. In this review, by summarizing current literature, it was shown that ginsenosides can modulate the fate of neurons and glia cells in different aspects, such as regulating cholinergic dysfunction, oxidative stress, apoptosis, and inflammation. Moreover, synaptic plasticity and neurogenesis can be promoted by the treatment of ginsenosides. The involved signaling pathways mainly include the PI3K/Akt, CREB/BDNF, Keap1/Nrf2 signaling, and NF-κB/NLRP3 inflammasome pathways. However, evidence of multiple targets of ginsenosides is inadequate, and the understanding of underlying mechanism remains limited. More work should be carried out to explore the exact targets of ginsenosides by novel techniques such as molecular docking, high-throughput screen, etc.

In addition, there are relatively few clinical trials of ginsenosides in improving CI, which may be related to the poor membrane permeability [29], low solubility [222], and low oral bioavailability of ginsenosides [223]. Recent studies have designed various ginsenoside delivery systems (e.g., polymeric microparticles, proliposome, and niosome) [20]. These delivery systems exert significant effects by improving water solubility, increasing the bioavailability, and enhancing the pharmacological activity of ginsenosides [224]. Therefore, it may provide a potential strategy for maximizing the potential in the clinical application of ginsenosides in the treatment of CI.

## Figures and Tables

**Figure 1 biomolecules-12-01310-f001:**
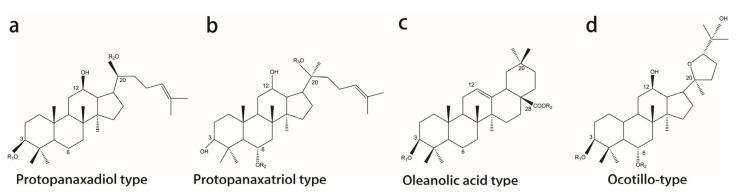
Chemical structures of ginsenosides. (**a**) Protopanaxadiol type; (**b**) protopanaxatriol type; (**c**) oleanolic acid type; and (**d**) ocotillo type.

**Figure 2 biomolecules-12-01310-f002:**
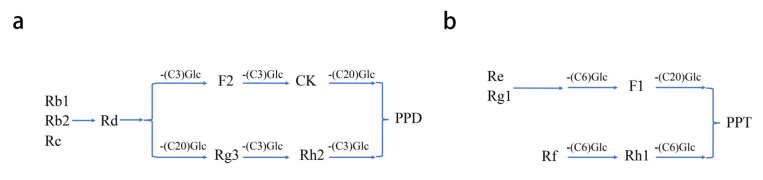
Metabolic pathway represents a deglycosylation process of PPD type (**a**) and PPT type (**b**) of ginsenosides.

**Figure 3 biomolecules-12-01310-f003:**
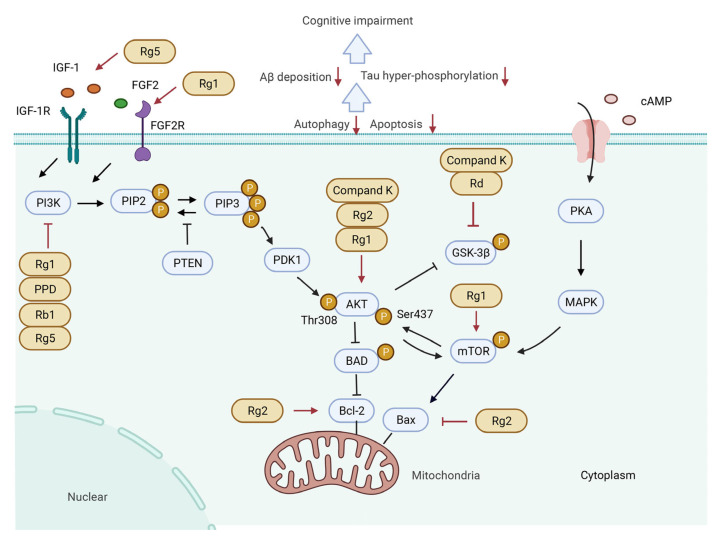
Molecular mechanisms of regulating PI3K/Akt signaling pathway by ginsenosides in treating cognitive impairment in the pathological model. (**1**) Rg5 and Rg1 activate PI3K/AKT pathway by stimulating IGF-1 and FGF2, perhaps eventually reducing Aβ deposition and tau hyperphosphorylation by inhibiting GSK-3β. Moreover, compound K, Rg2, and Rg1 can activate phosphorylation of AKT to reduce the content of GSK-3β. In addition, compound K and Rd directly attenuate GSK-3β to further inhibit Aβ deposition and tau hyperphosphorylation. (**2**) PI3K can be inhibited by Rg1, PPD, Rb1, and Rg5, further promoting the expression of anti-apoptotic protein Bcl-2, and increased the Bcl-2/Bax ratio to attenuate cell apoptosis. Additionally, Rg2 can directly promote the expression of Bcl-2. (**3**) Rg1 increases the expression of p-Akt and p-mTOR to inhibit mTOR-medicated autophagy.

**Figure 4 biomolecules-12-01310-f004:**
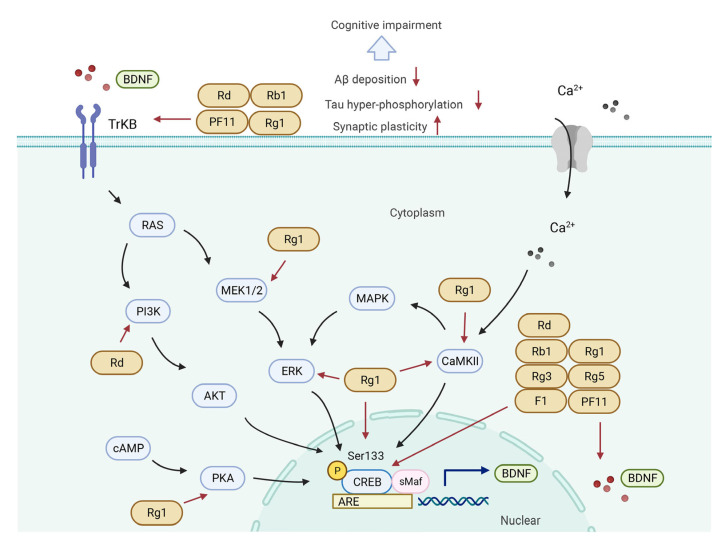
Molecular mechanisms of regulating CREB/BDNF signaling pathway by ginsenosides in treating cognitive impairment in the pathological model. CREB/BDNF signaling pathway can be activated by ginsenosides in multiple ways: (**1**) Rd, Rb1, PF 11, and Rg1 promote the expression of TrKB. (**2**) Rg1 and Rd increase the expression of MEK1/2 and PI3K, respectively. (**3**) Rg1 activates ERK, PKA, CREB, and CaMKII. (**3**) Rd, Rb1, Rg1, Rg3, Rg5, F1, and PF11 can promote both CREB and BDNF. All ginsenosides play a positive role in stimulating key factors in CREB/BDNF pathway to further improve synaptic plasticity and decrease Aβ deposition and tau hyperphosphorylation.

**Figure 5 biomolecules-12-01310-f005:**
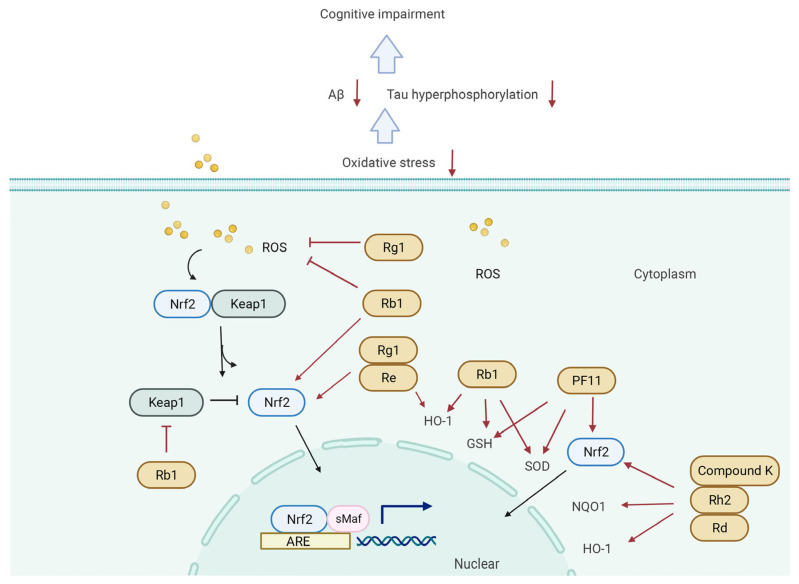
Molecular mechanisms of regulating Keap1/Nrf2 signaling pathway by ginsenosides in treating cognitive impairment in the pathological model. (**1**) Rg1 and Rb1 directly reduce the content of ROS in the cytoplasm. (**2**) Rb1, Rg1, Re, PF11, compound K, Rh2, and Rd promote the expression of Nrf2. (**3**) Rb1 inhibits the expression of Rb1. (**4**) Re and Rb1 promote the expression of HO-1. (**5**) Rb1 and PF11 increase the content of GSH; Rb1 and PF11 increase the content of SOD. All ginsenosides downregulate the oxidative stress in cells and further decrease Aβ deposition and tau hyperphosphorylation.

**Figure 6 biomolecules-12-01310-f006:**
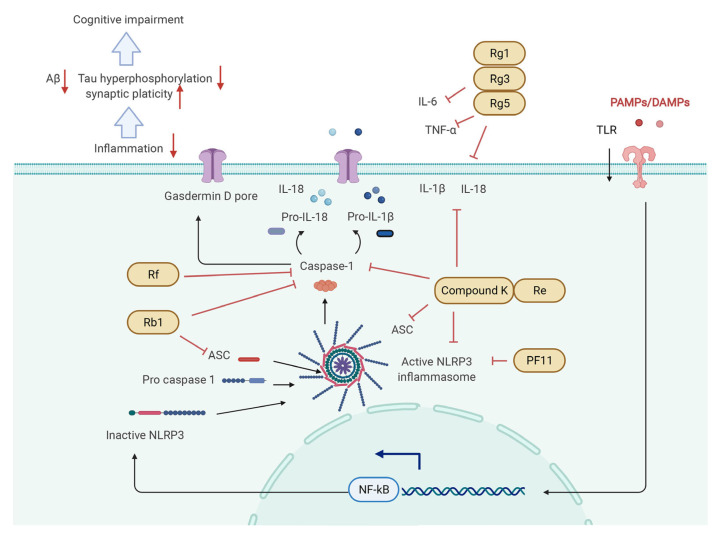
Molecular mechanisms of regulating NF-κB/NLRP3 pathway by ginsenosides in treating cognitive impairment in the pathological model. NF-κB pathway can be inhibited by ginsenosides via multiple ways: (**1**) Rf, Rb1, compound K, and Re inhibit the expression of caspase-1 to further reduce the level of IL-18 and IL-1β. (**2**) Rb1, compound K, and Re inhibit the expression of ASC, an important factor for formatting NLRP3 inflammasome. (**3**) Compound K, Re, and PF11 can directly suppress activated NLRP3 inflammasome. (**4**) Rg1, Rg3, Rg5, compound K, and Re downregulate the level of IL-6, TNF-α, IL-18, and IL-1β. All ginsenosides can attenuate the inflammatory state and further improve synaptic plasticity and decrease Aβ deposition and tau hyperphosphorylation.

**Table 1 biomolecules-12-01310-t001:** Summary of ginseng saponin carbohydrates.

Classification	Saponins	Formula	R1(C3)	R2(C6)
Protopanaxadiol type				
(PPD-type)	Rb1	C_54_H_92_O_23_	Glc-Glc	H
	Rb2	C_54_H_90_O_22_	Glc-Glc	H
	Rc	C_53_H_90_O_22_	Glc-Glc	H
	Rd	C_48_H_82_O_18_	Glc-Glc	H
	Rg3	C_42_H_72_O_13_	Glc-Glc	H
	Rh2	C_36_H_62_O_8_	Glc	H
	Compound K	C_36_H_62_O_8_	H	H
Protopanaxatriol type				
(PPT-type)	Rg1	C_42_H_72_O_14_	H	Glc
	Rg2	C_42_H_72_O_13_	H	Glc-Rha
	Rh1	C_36_H_62_O_9_	H	Glc
	Re	C_30_H_42_O_4_	H	Glc-Rha
	Rf	C_42_H_72_O_14_	H	Glc-Glc
	F1	C_36_H_62_O_9_	H	H
Oleanolic acid type				
	Ro	C_48_H_76_O_19_	GlcUA-Glc	Glc
Ocotillol-type				
	P-F11	C_42_H_72_O_14_	Glc-Rha	None

**Table 2 biomolecules-12-01310-t002:** Clinical trials of ginseng or ginsenosides in improving CI.

Treatment	Study Design	Diseases	Sample Size	Treatment Dosage and Rote	Outcomes	References
Sun ginseng-135 (ginsenoside complex)	Randomized, open-label trial	AD	40	From 1.5 g/day up to 4.5 g/day; oral administration	MMSE scores and ADAS scores were improved	Heo et al. [11]
Korean red ginseng (KRG) (containing 8.54% of ginsenosides)	Randomized, open-label trial	AD	31	4.5 g/day or 9 g/day; oral administration	CDR and ADAS scores were significantly improved	Heo et al. [44]
*Panax ginseng* powder (contains total 8.19% of ginsenosides)	Randomized controlled trial	AD	97	4.5 g/day or 9 g/day; oral administration	ADAS and the MMSE score show improvements during ginseng treatment	Lee et al. [45]
KRG	Randomized, double-blind, placebo-controlled trail	Healthy individuals	51	1000 mg/d (500 mg/capsule × 2 capsules). oral administration	Gray matter volume in the left parahippocampal gyrus and the composite score of combined cognitive function were significantly increased	Namgung et al. [46]
HT1001 (consists of a mixture of important ginsenosides)	Randomized controlled trial	Schizophrenia	64	100 mg/day; oral administration	Visual working memory was significantly improved, extrapyramidal symptoms were significantly reduced	Chen et al. [48]
KRG (contained major ginsenosides)	Double-blind, randomized, placebo-controlled trial	Participants with high-stress occupations	63	500 mg of KRG powder per capsule; oral administration	Triglyceride levels were significantly increased, epinephrine level was decreased	Beak et al. [49]
Cereboost™ (10.65% ginsenosides )	Randomized, double-blind, placebo-controlled crossover	Healthy young adults	32	100, 200, or 400 mg/day; oral administration	Working memory was improved	Scholey et al. [50]
G115 (ginsenoside complex)	Double-blind, placebo-controlled, balanced crossover	Healthy young adults	27	200 mg/day; oral administration	Enhancing cognitive performance	Reay et al. [51]

**Table 3 biomolecules-12-01310-t003:** Summary of anti-CI ginsenosides in experimental studies.

Compound	Diseases Model	Species	Administration Method and Duration	Treatment Dosage	Results	References
Rb1	AD	SAMP8 mice	Intragastric administration (8 weeks)	30 and 60 µmol/kg	Repaired neuronal cells loss and inhibited the activation of astrocyte and microglia in hippocampus	Yang et al. [53]
	AD	Aβ_1–40_-induced rat	Intraperitoneal injection (2 weeks)	12.5, 25, and 50 mg/kg	Inhibited the levels of pro-apoptosis mediators and improved the levels of anti-apoptosis mediators	Wang et al. [62]
	AD	Aβ_1–40_-induced rat	Intragastric administration (2 weeks)	12.5 mg/kg/d, 25.0 mg/kg/ d, and 50.0 mg/kg/d)	Altered the amyloidogenic process of APP into non-amyloidogenic process	Lin et al. [63]
	AD	ICR mice	Intragastric administration (4 months)	20 mg/kg/day	Protected against Al-induced toxicity	Zhao et al. [64]
	PD	C57BL/6 mice	Intraperitoneal injection	10 mg/kg	Enhanced GABA release	Liu et al. [66]
	PD	PC12 cells	-	50 or 150 μM	Reduced the cytotoxicity of MPTP	Rudakewich et al. [67]
	Focal cerebral ischemia	Sprague–Dawley (SD) rats	Intraperitoneal injection	100 mg/kg, 50 mg/kg, 25 mg/kg	Increased the expressions of P-Akt, P-mTOR, and reducedP-PTEN and caspase-3	Yan et al. [68]
	Diabetes	Primary hippocampal neuronal cells	-	1 µM	inhibited GSK3β-mediated CHOP induction	Liu et al. [69]
	Diabetes	C57BL/6N male mice	Intragastric administration (4 weeks)	30 mg/kg	Relieved glucose intolerance, inhibited Cdk5/p35	Yang et al. [70]
Rg1	AD	SAMP8 mice	Intragastric administration (8 weeks)	30 and 60 µmol/kg	Repaired neuronal cells loss and inhibited the activation of astrocyte and microglia in hippocampus	Yang et al. [53]
	AD	SAMP8 mice	Intragastric administration (3 months)	2.5, 5.0, and 10 mg/kg	Attenuated hippocampal Aβ content	Shi et al. [54]
	AD	SAMP8 mice	Intragastric administration	7.5 mg/kg/day	Suppressed neuron cell apoptosis	Shi et al. [55]
	AD	N2a-APP695 cells	-	2.5 μM	Decreased the levels of Aβ_1–40_ and Aβ_1–42_	Chen et al. [56]
	AD	SD rats	Intracerebroventricular injection (25 days)	20 mg/kg	Attenuated Aβ formation	Song et al [57]
	AD	APP/PS1 mice	Intragastric administration (6 to 9 months)	5 mg/kg	Decreased the p-Tau level, amyloid precursor protein (APP) expression, and Aβ generation	Zhang et al. [59]
	AD	Male conventional tree shrews	Intraperitoneally intragastric administration (8 weeks)	7.5, 15, and 30 mg/kg	Changed the abundance of gut microbiota	Wang et al. [60]
	AD	Tree shrews	Intracerebroventricular injection (6 weeks)	30 mg/kg/day	Altered the microbiota abundance, affected the expression of apoptosis proteins	Guo et al. [61]
	PD	PC12 cells	-	50 or 150 μM	Reduced the cytotoxicity of MPTP	Rudakewich et al. [67]
	Alcohol-induced psychomotor and cognitive deficits	ICR mice	Intragastric administration (14 days)	3 mg/kg, 6 mg/kg, and 12 mg/kg	Meliorated repeated alcohol-induced cognitive deficits	Huang et al. [71]
	Aging	SD rats	Subcutaneous injection (42 days)	20 mg/kg·d	Improved cognitive ability, protected NSCs/NPCs, and promoted neurogenesis	Zhu et al. [72]
	Aging	Kunming mice	Intraperitoneal injection (42 days)	10, 20 mg·kg^−1^	Inhibited apoptosis	Zhong et al. [73]
Rd	Transient forebrain ischemia	Primary neurons of SD rats	-	10 μM	Attenuated Tau protein phosphorylation	Zhang et al. [58]
	Chronic cerebral hypoperfusion	C57BL/6J mice	Intraperitoneal injection (21 days)	10 or 30 mg/kg	Upregulated BDNF and increased neuron survival	Wan et al. [74]
	Chronic restraint stress	C57BL/6J mice	Intragastric administration (7 days)	10, 20, or 40 mg/kg	Mitigated oxidative stress and inflammation	Wang et al. [75]
Compound K	AD	Mouse hippocampal HT22 cells/male C57BL/6J mice	Intragastric administration (2 weeks)	1, 5, 10 mg/kg	Reduced reactive oxygen species-	Seo et al. [65]
Rg5	Thermal stress	HT22 cells	-	20 mg/ mL and 40 mg/mL	Prevented apoptosis	Choi et al. [76]
Re	Diabetes	SD rats	Intragastric administration (8 weeks)	40 mg/kg	Attenuated diabetes-associated cognitive decline	Liu et al. [77]

**Table 4 biomolecules-12-01310-t004:** Other bioactive compounds from herbal plants contribute to prevention of CI.

Compounds	Herbal Plants	Diseases	Main Findings	References
Astragaloside IV	*Astmgali Radix.*	AD	Inhibited of microglial activation	Chen et al. [93]
EGb 761	*Ginkgo biloba*	AD	Neuroprotective effect	Mazza et al. [94]
Puerarin	*Pueraria lobata*	AD	Reduced impairment of iron metabolism	Yu et al. [95]
EGB761 and HBO	*Ginkgo biloba*	AD	Reduced cell toxicity and oxidative stress	Tian et al. [96]
4-O-methylhonokiol	*Magnolia officinalis*	AD	Attenuated β-amyloid-induced memory impairment	Lee et al. [97]
xiecaoside A–C and xiecaoline A, xiecaoside D, xiecaoside E	*Valeriana amurensis*	AD	Protected against Aβ-induced toxicity	Wang et al. [98]
Amurensin G, r-2-viniferin and trans-ε-viniferin	*Vitis amurensis*	AD	Protected against amyloid β protein (25–35)-induced neurotoxicity	Jeong et al. [99]
Dendrobium nobile Lindl. alkaloid	*Dendrobium nobile Lindl*	AD	Suppressed NLRP3-mediated pyroptosis	Li et al. [100]
Spinosin	*Zizyphus jujuba var. spinosa*	AD	Ameliorated Aβ_1–42_ oligomer-induced memory impairment	Ko et al. [101]

## Data Availability

Not applicable.

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
