# Peer review of "Ginsenoside and Its Therapeutic Potential for Cognitive Impairment"

_biomolecules, 2022, doi:10.3390/biom12091310_

Round 1

Reviewer 1 Report

I think this paper is interesting and can be considered for publication in Biomolecules, however several points should be revised before processing further.

1. The references do not follow Biomolecules style, please check carefully and revise throughout.

2. First letter of compound A and B in Figure 1 should be capitalized.

3. The authors described about the role of ginsenosides against CI (Cognititive Impairment) but please discuss more about what compounds among ginsenosides may play strong role against CI.

4. Further paper should be cited, for instance (J Ginseng Res. 2018 Oct; 42(4): 401–411.) and (Antioxidants (Basel). 2020 Mar; 9(3): 229.). 

5. There are many compounds have been described related to the prevention of CI, please tabulate and compare with ginsenosides.

6. Discuss about content of ginsenosides varied among Ginseng from China, Korea, Russia and other countries currently provide Ginseng.

7. Although ginsenosides are commonly described to be well tolerated among patients, but please provide more information and discuss about the side effects of ginsenosides, for instance  nervousness, insomnia, changes in blood pressure, breast pain, vaginal bleeding, vomiting, diarrhea, and mania.

Author Response

Dear Editors and Reviewers:

We are grateful to editors and reviewers for their efforts in reviewing our paper. All comments are very helpful to improve the quality of this paper. The manuscript has been carefully revised according to the reviewer’s suggestions.

A revised manuscript with corrections marked in red was provided for easy check/editing purposes, and our point-by-point responses are presented accordingly.

We appreciate the reviewer’s time and consideration. We sincerely hope this manuscript will be finally acceptable to be published on Biomolecules. The point-by-point responses to each comment were presented as follows:

  1. The references do not follow Biomolecules style, please check carefully and revise throughout.

Response: We would like to thank the reviewer very much for her/his thorough appraisal of our paper. All references have been checked and revised to match the Biomolecules style.

  1. First letter of compound A and B in Figure 1 should be capitalized.

Response: We would like to thank the reviewer very much for this important comment. The first letter of compound A and B in Figure 1 have been capitalized as reviewer suggested.

  1. The authors described about the role of ginsenosides against CI (Cognititive Impairment) but please discuss more about what compounds among ginsenosides may play strong role against CI.

Response:We would like to thank the reviewer very much for this valuable comment. According to reviewer’s suggestion, we have discussed more about the compounds among ginsenosides in the revised version as follows:

To conclude, multiple ginsenosides are able to ameliorate CI in different diseases. Among them, which ones may play stronger role against CI? Researches have demonstrated that a decrease in the number of sugar moieties of ginsenosides was shown to weaken their anti-CI effects. Several studies have compared the effects and mechanisms on improving CI of Rb1 and Rg1, which have been proved to be high in content of Panax ginseng. In scopolamine-induced mice, both Rg1 and Rb1 intraperitoneal administration at 6 and 12 mg/kg improved CI in mice and inhibited the decrease of 5-HT induced by scopolamine; However, Rb1, with four sugars, was more effective than ginsenoside Rg1, which contains only two sugars. Similar results have shown in Yang et al’s study. Moreover, another study reported that both Rg1 and Rg2 by intraperitoneal injection at 30 mg/kg/day improved spatial learning and memory deficit in APP/PS1 mice, but the effect of Rg1, is more obvious. PF11, an ocotillol-type ginsenosides, whose mother nucleus structure is similar to that of dammarane ginsenosides, also has strong effect on the prevention of CI, especially in the treatment of AD-related CI. Interestingly, compound K, a metabolite of PPD-type ginsenosides Rb1, Rb2, and Rc, seems more bioavailable than other ginsenosides such as Rd. However, at present, there are relatively few studies on the comparison of different effects of various ginsenosides on preventing CI, which may be an important perspective for future studies.

  1. Further paper should be cited, for instance (J Ginseng Res. 2018 Oct; 42(4): 401–411.) and (Antioxidants (Basel). 2020 Mar; 9(3): 229.).

Response: We would like to thank the reviewer very much for this important comment. The paper mentioned has been cited in the revised version.

  1. There are many compounds have been described related to the prevention of CI, please tabulate and compare with ginsenosides.

Response: We would like to thank the reviewer very much for this comment. We have created a table of several bioactive compounds from herbal plants that contribute to the prevention of CI in the revised version (Table 4).

  1. Discuss about content of ginsenosides varied among Ginseng from China, Korea, Russia and other countries currently provide Ginseng.

Response: We would like to thank the reviewer very much for this valuable comment. We have re-checked the related literature and revised several sentences. According to the literature, there are 3 major herbal plants called “ginseng”: Asian ginseng (Panax ginseng C.A. Meyer), American ginseng (Panax quinquefolius) and Notoginseng (Panax notoginseng). Actually, the “ginseng” from China, Korea, and Russia is the same species called Asian ginseng. We have discussed the content of ginsenosides varied among Asian ginseng, American ginseng, and Notoginseng. The added parts were as follows:

  1. Ginseng, the root of Panax ginsengA. Meyer, Araliaceae (also called Asian ginseng), is a perennial herb widely used in Traditional Chinese Medicine (TCM) mainly distributed in China, Korea, and Russia, and it was also dubbed the “the king of herbs”. The name “ginseng” is also applied to other herbal plants, e.g., American ginseng (Panax quinquefolius L.), Notoginseng (Panax notoginseng (Burk.) F. H. Chen).
  2. Different types of ginseng possess different content of ginsenosides. A study showed a total of 14 ginsenosides (Rg1, Re, Rf, Rh1, Rg2, Rb1, Rc, Rb2, Rb3, Rd, Rg3, Rk1, Rg5, and Rh2) were determined in Asian ginseng using high-performance liquid chromatography coupled with evaporative light scattering detection (HPLC-ELSD), while in American ginseng, 43 ginsenosides were identified. Besides, American ginseng has a different ginsenoside spectrum compared with Asian ginseng: 1) the top 3 major ginsenoside isolated from Asian ginseng are Rb1, Rg1, and Rb2, while the major content in American ginseng consists of Rb1, Re, and Rd. 2) PF11 only exist in American ginseng, while gin-senoside Rf is a special ingredient in Asian ginseng. For Notoginseng, it contains some of the same ginsenosides as Asian ginseng, e.g., Rb1, Rd, Re, Rg1, Rg2, Rh1, however, no ocotillol-type, and oleanane-type ginsenosides can be found in Notoginseng. In addition, some saponins are unique to notoginseng, e.g., notoginsenosides R1 and Rt.

  1. Although ginsenosides are commonly described to be well tolerated among patients, but please provide more information and discuss about the side effects of ginsenosides, for instance nervousness, insomnia, changes in blood pressure, breast pain, vaginal bleeding, vomiting, diarrhea, and mania.

Response: We would like to thank the reviewer very much for this valuable comment. According to the reviewer’s suggestion, the side effect of ginsenosides have been discussed in the revised version as follows:

In spite of the strong anti-CI effect of ginsenosides in CNS or Non-CNS diseases, the side effects may occur especially with long-term use, called “ginseng abuse syndrome”. The most common side effects of ginsenosides include diarrhea, vomiting, hypertension, skin rash, insomnia, breast pain, and vaginal bleeding. In a two-year human study, 14 to 26 of 133 participants showed symptoms of hypertension, euphoria, irritability, insomnia, edema, and diarrhea after taking ginseng for a long time. However, in the study, the validity of these studies is difficult to assess because of the absence of a control group and the fact that subjects used different ginseng preparations with the dosage ranging from 1 to 30g per day and were not controlled for other bioactive substances intake (e.g., caffeine). Because one of the main side effects of ginseng is hypertension, it is highly recommended that patients should discontinue ginseng use at least 7 days before surgery to reduce perioperative morbidity related to the herbal supplements. In addition, several studies also reported that ginseng may cause breast pain and vaginal bleeding in postmenopausal women in a few cases, which might be related to the physiological estrogen-like effect of ginseng. Therefore, the new administration routes related to novel materials (e.g., nanoparticles) that may potentially reduce the side effect of ginseng or ginsenosides treatment are urgently needed.

  1. The references do not follow Biomolecules style, please check carefully and revise throughout.

Response: We would like to thank the reviewer very much for her/his thorough appraisal of our paper. All references have been checked and revised to match the Biomolecules style.

  1. First letter of compound A and B in Figure 1 should be capitalized.

Response: We would like to thank the reviewer very much for this important comment. The first letter of compound A and B in Figure 1 have been capitalized as reviewer suggested.

  1. The authors described about the role of ginsenosides against CI (Cognititive Impairment) but please discuss more about what compounds among ginsenosides may play strong role against CI.

Response: We would like to thank the reviewer very much for this valuable comment. According to the reviewer’s suggestion, we have discussed more the compounds among ginsenosides in the revised version as follows:

To conclude, multiple ginsenosides are able to ameliorate CI in different diseases. Among them, which ones may play a stronger role against CI? Research has demonstrated that a decrease in the number of sugar moieties of ginsenosides was shown to weaken their anti-CI effects. Several studies have compared the effects and mechanisms of improving CI of Rb1 and Rg1, which have been proved to be high in content of Panax ginseng. In scopolamine-induced mice, both Rg1 and Rb1 intraperitoneal administration at 6 and 12 mg/kg improved CI in mice and inhibited the decrease of 5-HT induced by scopolamine; However, Rb1, with four sugars, was more effective than ginsenoside Rg1, which contains only two sugars. Similar results have been shown in Yang et al’s study. Moreover, another study reported that both Rg1 and Rg2 by intraperitoneal injection at 30 mg/kg/day improved spatial learning and memory deficit in APP/PS1 mice, but the effect of Rg1, is more obvious. PF11, ocotillol-type ginsenosides, whose mother nucleus structure is similar to that of dammarane ginsenosides, also has a strong effect on the prevention of CI, especially in the treatment of AD-related CI. Interestingly, compound K, a metabolite of PPD-type ginsenosides Rb1, Rb2, and Rc, seems more bioavailable than other ginsenosides such as Rd. However, at present, there are relatively few studies on the comparison of different effects of various ginsenosides on preventing CI, which may be an important perspective for future studies.

  1. Further paper should be cited, for instance (J Ginseng Res. 2018 Oct; 42(4): 401–411.) and (Antioxidants (Basel). 2020 Mar; 9(3): 229.).

Response: We would like to thank the reviewer very much for this important comment. The paper mentioned has been cited in the revised version.

  1. There are many compounds that have been described related to the prevention of CI, please tabulate and compare with ginsenosides.

Response: We would like to thank the reviewer very much for this comment. We have created a table of several bioactive compounds from herbal plants that contribute to the prevention of CI in the revised version (Table 4).

  1. Discuss about content of ginsenosides varied among Ginseng from China, Korea, Russia and other countries currently provide Ginseng.

Response: We would like to thank the reviewer very much for this valuable comment. We have re-checked the related literature and revised several sentences. According to the literature, there are 3 major herbal plants called “ginseng”: Asian ginseng (Panax ginseng C.A. Meyer), American ginseng (Panax quinquefolius) and Notoginseng (Panax notoginseng). Actually, the “ginseng” from China, Korea, and Russia is the same species called Asian ginseng. We have discussed the content of ginsenosides varied among Asian ginseng, American ginseng and Notoginseng. The added parts were as follows:

  1. Ginseng, the root of Panax ginsengA. Meyer, Araliaceae (also called Asian ginseng), is a perennial herb widely used in Traditional Chinese Medicine (TCM) mainly distributed in China, Korea, and Russia, and it was also dubbed the “the king of herbs”. The name “ginseng” is also applied to other herbal plants, e.g., American ginseng (Panax quinquefolius L.), Notoginseng (Panax notoginseng (Burk.) F. H. Chen).
  2. Different types of ginseng possess different content of ginsenosides. A study showed a total of 14 ginsenosides (Rg1, Re, Rf, Rh1, Rg2, Rb1, Rc, Rb2, Rb3, Rd, Rg3, Rk1, Rg5, and Rh2) were determined in Asian ginseng using high-performance liquid chromatography coupled with evaporative light scattering detection (HPLC-ELSD), while in American ginseng, 43 ginsenosides were identified. Besides, American ginseng has a different ginsenoside spectrum compared with Asian ginseng: 1) the top 3 major ginsenosides isolated from Asian ginseng are Rb1, Rg1, and Rb2, while the major content in American ginseng consists of Rb1, Re, and Rd. 2) PF11 only exist in American ginseng, while ginsenoside Rf is a special ingredient in Asian ginseng. For Notoginseng, it contains some of the same ginsenosides as Asian ginseng, e.g., Rb1, Rd, Re, Rg1, Rg2, Rh1, however, no ocotillol-type, and oleanane-type ginsenosides can be found in Notoginseng. In addition, some saponins are unique to notoginseng, e.g., notoginsenosides R1 and Rt.

  1. Although ginsenosides are commonly described to be well tolerated among patients, but please provide more information and discuss about the side effects of ginsenosides, for instance nervousness, insomnia, changes in blood pressure, breast pain, vaginal bleeding, vomiting, diarrhea, and mania.

Response: We would like to thank the reviewer very much for this valuable comment. According to reviewer’s suggestion, the side effect of ginsenosides have been discussed in the revised version as follows:

In spite of the strong anti-CI effect of ginsenosides in CNS or Non-CNS diseases, the side effects may occur especially with long-term use, called “ginseng abuse syndrome”. The most common side effects of ginsenosides include diarrhea, vomiting, hypertension, skin rash, insomnia, breast pain, and vaginal bleeding. In a two-year human study, 14 to 26 of 133 participants showed symptoms of hypertension, euphoria, irritability, insomnia, edema and diarrhea after taking ginseng for a long time. However, in the study, the validity of these studies is difficult to assess because of the absence of a control group and the fact that subjects used different ginseng preparations with the dosage ranging from 1 to 30g per day and were not controlled for other bioactive substances intake (e.g., caffeine). Because one of the main side effects of ginseng is hypertension, it is highly recommended that patients should discontinue ginseng use at least 7 days before surgery to reduce perioperative morbidity related to the herbal supplements. In addition, several studies also reported that ginseng may cause breast pain and vaginal bleeding in postmenopausal women in a few cases, which might be related to the physiological estrogen-like effect of ginseng. Therefore, the new administration routes related to novel materials (e.g., nanoparticles) that may potentially reduce the side effect of ginseng or ginsenosides treatment are urgently needed.

Reviewer 2 Report

This is very interesting review paper and helpful readers to know related field and recent trend. I have several points which should be added as below.

- a lot of type errors for symbolic words (eg., L564, 575 etc...) should be corrected.

-Scientific name (eg., Panax ginseng) should be italic.

- Reference section needs extensive correction (please check journal's policy for reference section). Also paper title should be lowercase.

- Authors should include Pharmacokinetic aspect of ginsenosides particularly in brain (eg., penetration of ginsenosides and distributed ginsenosides in brain etc.).

-Title of Fig. 5 should be changed. it is same with Fig. 2.

-Also discuss structural aspect of ginsenosides between PPD type and PPT types in this pharmacology. 

- "Table 2. Clinical trials of ginsenosides in improving cognitive function" should be changed, since KRG data are also included in this table.

- Another Table including ginsenosides and their in vitro and in vivo activities, experimental conditions, and cell or mouse names, etc, should be also prepared.

Author Response

Dear Editors and Reviewers:

We are grateful to editors and reviewers for their efforts in reviewing our paper. All comments are very helpful to improve the quality of this paper. The manuscript has been carefully revised according to the reviewer’s suggestions.

A revised manuscript with corrections marked in red was provided for easy check/editing purposes, and our point-by-point responses are presented accordingly.

We appreciate the reviewer’s time and consideration. We sincerely hope this manuscript will be finally acceptable to be published on Biomolecules. The point-by-point responses to each comment were as follows:

- a lot of type errors for symbolic words (eg., L564, 575 etc...) should be corrected.

Response: We would like to thank the reviewer very much for her/his thorough appraisal of our paper. The symbolic words have been corrected in the revised version.

-Scientific name (eg., Panax ginseng) should be italic.

Response: We would like to thank the reviewer very much for this important comment. The scientific name was revised as suggested.

- Reference section needs extensive correction (please check journal's policy for reference section). Also paper title should be lowercase.

Response: We would like to thank the reviewer very much for this comment. The references and title have been revised in order to match journal's policy.

- Authors should include Pharmacokinetic aspect of ginsenosides particularly in brain (eg., penetration of ginsenosides and distributed ginsenosides in brain etc.).

Response: We would like to thank the reviewer very much for this valuable comment. We have discussed the pharmacokinetic aspect of ginsenosides in brain as suggested. The added part was as follows:

Several studies have reported the pharmacokinetics of ginsenosides in brain tissue. One study showed that ginsenosides Rb1, Rb2, Rc, Rd, Re, Rf, Rg1, Rg3, and Ro were rap-idly transported into the brain at 5 min after intravenous administration of ShenMai Injection, a TCM preparation mainly composed of ginseng extract. In addition, ginsenosides Rb1, Rg1, Ro, and Re can be detected in rat brain tissue after oral administration of Jia-Wei-Qi-Fu-Yin, a TCM decoction with ginseng as the main gradient. These data indicate that ginsenosides could penetrate through the blood–brain barrier (BBB). However, another study found that the average brain concentration of ginsenosides Rb1, Rg1, and Re was 8 to 15 times lower than the corresponding content in plasma after oral administration of ginseng extract in rats, which indicated they have poor permeability to BBB. There are several studies focused on the distribution of ginsenosides in different regions of brain tissue. The imaging result of the brain implied that Rg1 might be distributed in the pons and medulla oblongata region of the brain at 15 min after intravenous administration in rats. In addition, Re can be rapidly distributed into cerebrospinal fluid and showed linear pharmacokinetics after subcutaneous injection in rats. Moreover, ginsenosides Rg1, Rb1, Re, and Rd could be detected in the hippocampus, hypothalamus, olfactory bulb, striatum, cortex, and medulla oblongata of rats after administration of Panax notoginseng saponins through the nostril. However, the knowledge of BBB permeability and distribution of ginsenosides in brain tissue remains limited, a better understanding of the pharmacokinetics of ginsenosides in the brain might contribute substantially to further research of their functions and mechanisms.

-Title of Fig. 5 should be changed. it is same with Fig. 2.

Response: We would like to thank the reviewer very much for this comment. The title of Fig. 5 has been checked and revised.

-Also discuss structural aspect of ginsenosides between PPD type and PPT types in this pharmacology.

Response: We would like to thank the reviewer very much for this valuable comment. Based on the reviewer’s suggestion, the structural aspect of ginsenosides between PPD type and PPT types in this pharmacology has been discussed in the revised version. The added part was as follows:

The difference in pharmacokinetic features between PPD- and PPT-type ginsenosides is closely related to the number, type and position of sugar chain linked to the sapogenin moiety. As shown in Figure 2, metabolic pathway of PPD-type 20(S)-ginsenoside or PPT-type 20(S)-ginsenoside both have two deglycosylation process in intestine, metabolic pathway represents deglycosylation at C3, C6, or C20 position by β-glucosidase from in-testinal microbiota. Through stepwise deglycosylation, the ginsenosides Rd is transformed into F2 and Rg3, while Rg1, Re and Rf are transformed into F1 and Rh1, respectively. Moreover, F2 and Rg3 further become compound K and Rh2, respectively. The final deglycosylated metabolites are presented as 20(S)-PPD and 20(S)-PPT.

- "Table 2. Clinical trials of ginsenosides in improving cognitive function" should be changed, since KRG data are also included in this table.

Response: We would like to thank the reviewer very much for this comment. We have corrected the name of the tables as “Clinical trials of ginseng or ginsenosides in improving CI” according to reviewer’s suggestion.

- Another Table including ginsenosides and their in vitro and in vivo activities, experimental conditions, and cell or mouse names, etc, should be also prepared.

Response: We would like to thank the reviewer very much for this valuable comment. We have prepared the table (Table 3) as the reviewer suggested. 

Round 2

Reviewer 1 Report

This paper has been revised in much better form. It can be published now after minor English revision. Thank you for your hard work.

Author Response

We would like to thank the reviewer very much for this valuable comment. 

Reviewer 2 Report

Authors have fully address all my issues. So, this paper is now acceptable without further changes.